# The GA4GH Variation Representation Specification: A computational framework for variation representation and federated identification

## Graphical abstract

## Authors

Alex H. Wagner, Lawrence Babb, Gil Alterovitz, ..., Andrew D. Yates, Robert R. Freimuth, Reece K. Hart

## Correspondence

alex.wagner@nationwidechildrens.org (A.H.W.), lbabb@broadinstitute.org (L.B.), reece@harts.net (R.K.H.)

## In brief

Wagner et al. report the Variation Representation Specification (VRS; pronounced "verse"), an open-source standard of the Global Alliance for Genomics and Health (GA4GH). VRS is an expressive computational framework for genomic (and other -omic) variation. The specification introduces a mechanism for the computable identification of variation, supporting federated data exchange.

## Highlights

- Introducing the Variation Representation Specification (VRS; pronounced "verse")

- VRS is a computational framework for representing biomolecular variation

- VRS enables computable identification of variation supporting federated data exchange

- VRS continues to evolve as an open-source, community-driven standard of the GA4GH

 Wagner et al., 2021, Cell Genomics 1, 100027
November 10, 2021 © 2021 The Authors.

# Cell Genomics

CellPress

## Technology

# The GA4GH Variation Representation Specification: A computational framework for variation representation and federated identification

Alex H. Wagner,[1,2,25,*] Lawrence Babb,[3,*] Gil Alterovitz,[4,5] Michael Baudis,[6] Matthew Brush,[7] Daniel L. Cameron,[8,9] Melissa Cline,[10] Malachi Griffith,[11] Obi L. Griffith,[11] Sarah E. Hunt,[12] David Kreda,[13] Jennifer M. Lee,[14] Stephanie Li,[15] Javier Lopez,[16] Eric Moyer,[17] Tristan Nelson,[18] Ronak Y. Patel,[19] Kevin Riehle,[19] Peter N. Robinson,[20] Shawn Rynearson,[21] Helen Schuilenburg,[12] Kirill Tsukanov,[12] Brian Walsh,[7] Melissa Konopko,[15] Heidi L. Rehm,[3,22] Andrew D. Yates,[12] Robert R. Freimuth,[23] and Reece K. Hart[3,24,*]

[1]Department of Pediatrics, The Ohio State University College of Medicine, Columbus, OH 43210, USA
[2]The Steve and Cindy Rasmussen Institute for Genomic Medicine, Nationwide Children's Hospital, Columbus, OH 43215, USA
[3]Medical and Population Genetics, Broad Institute of MIT and Harvard, Cambridge, MA 02142, USA
[4]Harvard Medical School, Boston, MA 02115, USA
[5]Department of Medicine, Brigham and Women's Hospital, Boston, MA 02115, USA
[6]University of Zurich and Swiss Institute of Bioinformatics, Zurich, Switzerland
[7]Oregon Health & Science University, Portland, OR 97239, USA
[8]Bioinformatics Division, Walter and Eliza Hall Institute of Medical Research, Melbourne, VIC, Australia
[9]Department of Medical Biology, University of Melbourne, Melbourne, VIC, Australia
[10]UC Santa Cruz Genomics Institute, Santa Cruz, CA 95060, USA
[11]Washington University School of Medicine, St. Louis, MO 63108, USA
[12]European Molecular Biology Laboratory, European Bioinformatics Institute, Wellcome Genome Campus, Hinxton, Cambridge CB10 1SD, UK
[13]Department of Biomedical Informatics, Harvard Medical School, Boston MA 02115, USA
[14]Essex Management LLC and National Cancer Institute, Rockville, MD 20850, USA
[15]The Global Alliance for Genomics and Health, Toronto, ON, Canada
[16]Genomics England, London EC1M 6BQ, UK
[17]National Center for Biotechnology Information, National Library of Medicine National Institutes of Health, Bethesda, MD 20894, USA
[18]Geisinger Health, Danville, PA 17822, USA
[19]Baylor College of Medicine, Houston, TX 77030, USA
[20]Jackson Laboratory for Genomic Medicine, Farmington, CT 06032, USA
[21]Utah Center for Genetic Discovery, University of Utah, Salt Lake City, UT 84112, USA
[22]Center for Genomic Medicine, Massachusetts General Hospital, Cambridge, MA 02142, USA
[23]Center for Individualized Medicine, Department of Artificial Intelligence and Informatics, Mayo Clinic, Rochester, MN 55905, USA
[24]MyOme, Inc., Menlo Park, CA 94070, USA
[25]Lead Contact
*Correspondence: alex.wagner@nationwidechildrens.org (A.H.W.), lbabb@broadinstitute.org (L.B.), reece@harts.net (R.K.H.)

## SUMMARY

Maximizing the personal, public, research, and clinical value of genomic information will require the reliable exchange of genetic variation data. We report here the Variation Representation Specification (VRS, pronounced "verse"), an extensible framework for the computable representation of variation that complements contemporary human-readable and flat file standards for genomic variation representation. VRS provides semantically precise representations of variation and leverages this design to enable federated identification of biomolecular variation with globally consistent and unique computed identifiers. The VRS framework includes a terminology and information model, machine-readable schema, data sharing conventions, and a reference implementation, each of which is intended to be broadly useful and freely available for community use. VRS was developed by a partnership among national information resource providers, public initiatives, and diagnostic testing laboratories under the auspices of the Global Alliance for Genomics and Health (GA4GH).

## INTRODUCTION

Precision medicine and contemporary biomedical research are increasingly driven by large, coordinated genome-guided efforts.[1–12] The analysis of patient genomic data in the clinical setting has enabled tremendous advances in health care delivery through genome-guided diagnosis and clinical decision support.[13–16] However, numerous technical, financial, and legal obstacles impede the adoption of genomic science on a global scale. To address these challenges, the Global Alliance for

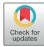

<div style="border:1px solid #a03030; padding:10px;">

**Box 1. The GA4GH Variation Representation Specification (VRS)**

VRS (pronounced "verse") is a precise and expressive specification for representing biomolecular variation. Here, we present the key concepts that collectively make VRS different from other genomic variant specifications and tools.

(1) Computational representation: VRS is about how computers describe and exchange data about molecular variation. A carefully defined terminology and inter-connecting data structures allow for computationally precise representation of variation that is fully computable.

(2) Federated identification: VRS was designed to derive identifiers from the nature of the variation itself; the same variant on a matching RefSeq and Ensembl transcripts looks different in human-readable nomenclatures but create the same computed identifier in VRS! This makes it easier to find the same variant across many independent databases.

(3) Expressivity: the modular nature of VRS objects makes it straightforward to represent both simple and complex variation concepts. Multiple types of biomolecular variation are already supported by VRS, including genomic, cytogenomic, transcript, and protein sequences, and at both the molecular and systemic level. This expressivity also makes VRS messages less suited to direct reading by people than some other variant specifications.

(4) Context precision: VRS does not lump together or link variation contexts; a variation can occur on only one sequence context (e.g., a GRCh38 chromosome) and is distinct from an analogous change on an aligned sequence context (e.g., a region from a GRCh37 chromosome or a projected transcript). This makes VRS well-suited to unambiguous representation but also leaves open challenges for linking contexts.

</div>

Genomics and Health (GA4GH) was formed as a policy-setting and technical standards development organization, and now comprises over 600 leading organizations in the domains of healthcare, research, patient advocacy, life science, and information technology.[17,18] GA4GH brings together expertise from a diverse international set of real-world, genomic-data-sharing Driver Projects. These Driver Projects contribute to domain-specific teams, or Work Streams, to promote the sharing of health and genomic data according to the findable, accessible, interoperable, and reusable (FAIR) principles.[18,19]

Ensuring that precision genomic medicine is effective for individuals and for health systems will require that clinicians, researchers, and testing laboratories communicate genomic variation and related information reliably. Although widely adopted standards for certain classes of variation already exist, many of these formats have been purpose-built for specific applications, including human-readable standards such as the Human Genome Variation Society (HGVS) variant nomenclature,[20] the International System for Human Cytogenomic Nomenclature (ISCN),[21] and the PharmVar Pharmacogenetics nomenclature,[22] as well as genome-oriented flat file formats such as the Variant Call Format (VCF),[23] among others (Table S1). All current standards have design constraints that preclude a comprehensive coverage of variation types and extensibility to new types.

In response to this need, the GA4GH Genomic Knowledge Standards (GKS) Work Stream led the development of the Variation Representation Specification (VRS, pronounced "verse"; https://vrs.ga4gh.org), a community-driven and extensible specification to standardize the exchange of diverse variation data. Throughout the specification and this manuscript, we use the term "variation" to mean the molecular or quantitative state of a referenced biological sequence. This very broad definition is in contrast with the commonly used term "variant," which refers to an alternative sequence relative to a reference sequence. This distinction sets the scope for VRS, which complements existing variant representation standards and provides an expressive framework for computational representation of variation. While many current standards (e.g., HGVS or ISCN nomenclatures) are designed to be amenable to visual interpretation by humans, VRS focuses on computational precision, expressiveness, and extensibility rather than human readability. As a result, VRS is more verbose than other contemporary human-readable variant representations but better suited to expressing complex variation concepts and adapting new classes of variation. VRS is a natural complement of human-readable nomenclatures when used for the exchange of genomic information from databases, clinical reports, or scientific manuscripts. VRS currently covers many classes of variation that are defined on a contiguous molecule such as single-nucleotide variants (SNVs), multi-nucleotide variants (MNVs), indels, repeats, and haplotypes. Collectively, these types of variation are called "molecular variation." In addition, VRS provides "systemic variation classes," which represent aggregated variation within a genome, cell, or other system instead of within a specific molecule. Systemic variation concepts include gene product expression, copy number variation, and genotypes (the set of alleles or haplotypes at a locus). The development of VRS has focused on those types of variation of greatest impact to the biomedical research and clinical genomics communities. The concepts in VRS are readily usable for description of variation from organisms with linear chromosome assemblies (Box 1).

As we move toward federated clinical genomics resources,[24] the development of a standard for expressive computational variation representation is increasingly important. as the number of systems describing variation increases, so too does the diversity of reference annotations and registered identifiers for describing variation. This growth has underscored the need for a global standard for describing and identifying variation. Here we describe the components of VRS and their use in enabling a federated system of resources for the functional and clinical annotation of variation. We summarize the key components enabling the precise and extensible representation of variation with VRS, including the underlying terminology, information model, schema, and conventions for computing globally consistent identifiers.

CellPress

**Figure 1. Components and extensions of the Variation Representation Specification**
VRS is a specification comprising multiple layered components (solid border blue boxes) that each serve as a foundation for the components above it. While VRS provides a full end-to-end framework, each component of the framework can be extended by the community into alternate forms (dash border gray boxes). For example, VRS provides a JSON Schema implementation based upon the Terminology and Information Model, but that same Terminology and Information Model may be used to build schemas in DTD, XSD, Google Protocol Buffers, Apache Thrift, or other data validation frameworks. This modular construction of VRS encourages interoperability across many scenarios and communities.

## DESIGN

To achieve a precise and computable representation of variation, VRS was designed as several interdependent components, including a terminology and information model, machine-readable schema, sharing conventions, and globally consistent and unique computed identifiers (Figure 1). These components allow for the specification to address multiple use cases and evolve to suit the needs of the community, and are maintained publicly online at https://vrs.ga4gh.org.

In totality, VRS provides a framework for the unique and semantically precise characterization of variation. To achieve this, VRS objects were designed as "value objects"—minimal information objects intrinsically defined by their content (e.g., a C > T transition in residue 2 of the sequence CCTAC)—rather than records for human presentation from system identifiers (e.g., clinvar:13961). This design choice is fundamental to each component of the specification; objects are immutable and have a specific meaning prescribed by the attributes they contain. Values extraneous to the meaning of a VRS object, such as the name of a location ("chr 7") or a label for a well-known variant ("BRAF V600E") are not captured directly within VRS objects, nor allowed to be added as user-defined fields within objects. This design ensures variation concepts are consistent in their construction and enables creation of globally consistent identifiers.

### Scope of VRS and other GA4GH genomic knowledge standards

There are many forms of variation that are assayed and annotated for use in clinical practice. These variation forms are shaped by the biological phenomena they describe, the assays used to measure those phenomena, and the methods used to transform the assay measurements into variation (Figure S1). The primary objective of VRS is to provide a framework for reliably expressing information about these many forms of variation for use in exchange between computer systems.

As value objects, VRS objects contain only the salient information for an instance of variation. VRS objects do not contain conventional transcript and chromosome identifiers, HGVS variant descriptions, gene names, reference sequence (which is implied by the location of the variation), inter-object relationships, or any other form of annotation or interpretation. Instead, the representation of annotations and interpretations is delegated to implementations. For example, different implementations may wish to implement local preferences for Ensembl, LRG, and RefSeq transcripts. Indeed, a key design goal for VRS is to decouple the intrinsic representation of variation from the local decisions about how variation is presented to humans. To standardize the interface between computational representations of variation and human presentations, the GKS Work Stream is developing the VRS Added Tools for Interoperable Loquacious Exchange (VRSATILE; pronounced "versatile") framework (https://vrsatile.readthedocs.io/).

Many public variant registries (e.g., CIViC,[25] ClinGen Allele Registry,[11] COSMIC,[6] and ClinVar[3]) aggregate multiple different variant contexts under controlled identifiers. For example, a genomic variant on multiple genome assemblies, the associated transcript changes, and predicted protein changes are each variant contexts that may all be linked under a single "variant" ID in such resources. As with local preferences for transcript selection, VRSATILE enables implementations to apply local policies regarding the use and presentation of aggregated sets of variation.

VRS does not support the characterization or capture of attributes describing experimental conditions. This means that VRS does not specify how to represent characteristics of the assay (e.g., microarray, whole-genome sequencing, gel electrophoresis), measurement (e.g., read count, signal intensity), or sample characteristics (e.g., sample ploidy, clonality, purity) beyond the variation itself. To support these cases, alongside other statements of biological and clinical evidence, the GKS Work Stream is building a larger framework to support variation annotations. The Variation Annotation Specification ("VA Spec"; https://github.com/ga4gh/va-spec) is being developed to closely integrate with developments in VRS and the components of the VRSATILE framework.

**Table 1. VRS Objects**

| Object type | Identifier prefix | VRS release | Purpose |
|---|---|---|---|
| **Variation** | | | |
| Allele | VA | v1.0 | contiguous insertions and deletions at a specific Location |
| Text | VT | v1.0 | other forms of variation (technical compatibility) |
| VariationSet | VS | v1.1 | collections of variation |
| Haplotype | VH | v1.1 | phased molecular variation |
| CopyNumber | VCN | v1.2 | an absolute systemic quantity/copy number variation |
| Gene Fusion | | planned | concept for representing deregulated or novel transcripts resulting from gene fusion |
| Gene Expression | | planned | systemic variation concept for representing gene product abundance |
| Genotype | | planned | summary of variation corresponding to a Location |
| Structural Variation | | planned | novel molecules composed from sequence originating at non-contiguous locations |
| **Location** | | | |
| SequenceLocation | VSL | v1.0 | locations on defined IUPAC character sequences |
| ChromosomeLocation | VCL | v1.1 | locations on defined cytogenetic regions |
| **SequenceExpression** | | | |
| LiteralSequenceExpression | N/A | v1.2 | used to define a specific sequence of literal IUPAC characters |
| DerivedSequenceExpression | N/A | v1.2 | used to define a sequence described by a sequence location |
| RepeatedSequenceExpression | N/A | v1.2 | used to define a specific sequence by a number of repeats of an indicated subsequence |
| **Interval** | | | |
| CytobandInterval | N/A | v1.1 | a cytogenetic interval specified using cytoband nomenclature |
| SequenceInterval | N/A | v1.2 | an interval on a sequence specified by a start and end coordinate or coordinate range. |

### Technical foundations and architectural design

To accomplish our objectives of a precise specification for representing the diversity of supported and planned variation concepts (Table 1), we have developed VRS to be extensible by designing minimal, standard building blocks that may be used interchangeably. For example, an Allele may be constructed using objects describing the location of the Allele on a sequence (SequenceLocation) or a chromosomal region (ChromosomeLocation), and with a molecular state described by one of several alternate Sequence Expressions (Figure S2). Sequence Expressions may also be used as a subject of Copy Number Variation. Future forms of variation, such as gene fusions, may use these same building blocks in new constructs. Sequence Expressions are extensible and may be expanded in the future to other forms as needed to support community cases.

VRS uses an adaptation of Semantic Versioning 2.0 (sem-ver.org) to distinguish releases that contain new backward-*compatible* features (minor version) from releases that contain backward-*incompatible* changes (major version). The first major version release of VRS, VRS 1.0, supported the Allele Variation class and related types; subsequent minor version releases introduced additional types of Variation and classes to support them (Table 1), all of which were backward compatible with prior versions. If backward-incompatible changes are required to improve the specification, they will be released in the next major version release (i.e., version 2.0) along with tooling to translate from version 1.x objects to version 2.0.

While VRS currently covers many common-use cases and continues to evolve (Table 1), practitioners may have a need for new classes of variation that are not (yet) supported by VRS. To meet this need, VRS provides a catch-all Text Variation class. This class allows implementers to use text to describe a variation and generate a computed identifier, thereby treating

novel forms of variation within the same framework as supported classes of variation. Because of the potential for abuse or misuse of this class, the specification does not require that implementations support Text Variation. For Text Variation that is eventually supported, VRS will support the notion of upgrading specific instances of unparsed Text Variation to structured forms of the same variation.

## RESULTS

### Terminology and information model

The VRS terminology and information model, informed by community authorities such as the ISCN and Sequence Ontology,[26] is the foundation for the VRS Schema. Definitions of biological terms in the scientific community may be abstract or intentionally ambiguous, reflecting imprecise or uncertain measures because of limitations in our understanding of those concepts. Occasionally, this creates divergent usage of terms across communities. For example, the term "genotype" has common meanings as either the alleles at a single genetic locus or as a collection of alleles at several genetic loci. While humans can readily discern between these overloaded definitions from context, abstract and ambiguous terms are not readily translatable into a computable representation of knowledge. Therefore, VRS begins with precise computational definitions for biological concepts that are essential to representing biomolecular variation. The VRS information model specifies how the computational definitions are represented semantically as inter-related objects and how values are to be represented in fields.

An important distinction made in VRS is between variation on a single resultant molecule ("molecular variation") from variation that pertains to an aggregated observation of many molecules ("systemic variation"). Molecular variation includes substitutions, insertions, repeats, tandem duplications, deletions, haplotypes, and structural rearrangements. Systemic variation, in contrast, is used to describe gene expression variation, copy number gain/ loss variation, and genotypes (planned) that may involve several molecules within a system (e.g., genome). For example, the HGVS expression "NC_000001.10:g.(?_15764950)(15765020_?) dup" has been used by some clinical labs to describe a copy number gain, assayed by a method that cannot confirm that the gain is a result of a tandem duplication. However, the use of the dup syntax, by definition in the HGVS recommendations, characterizes this event as a tandem duplication. These two variation concepts—a tandem duplication on a molecule and a systemic copy number gain—are often described ambiguously, leading to potential misinterpretation of descriptions of variation by data consumers. Importantly, VRS also maintains the separation of concerns between these two statement types; if a data source intends to specify a tandem duplication *and* specify a systemic copy number gain as a result of that duplication, these are represented as two distinct variation objects associated with the corresponding sample by that source. Wide adoption of VRS will enable systems to appropriately distinguish molecular and systemic copy number changes, from which derived human-readable forms (e.g. a CNV representation or a HGVS dup) may be generated.

VRS enables genomic data providers to clearly delineate these two types of variation and several other common variation con-

cepts. This is enabled through precise technical definitions of classes for Molecular Variation (Allele, Haplotype) and Systemic Variation (Copy Number) concepts. Additional technical Variation concepts (Variation Set, Text) and concepts supporting these variation types (Location, Sequence Expression, and Interval) are also defined. Text variation serves as a useful mechanism for an interim representation of variants that are otherwise not yet representable with VRS. Some examples of these are covered in our planned concepts under active development, including structural variation, genotypes, and transcript locations (Table 1). Technical definitions for primitive concepts supporting these objects (CURIE [Compact Uniform Resource Identifier], Residue, Sequence, Number, Definite Range, Indefinite Range) are also provided.

Variation, Location, Sequence Expression, and Interval are all extensible abstract base classes (Figure 2). This feature of the specification enables extensibility of the model, such as the reuse of Location classes to describe not only sequence variation, but also variation described by a gene concept or a cytoband. Similarly, while Alleles will often be composed from a Sequence Location (a location defined by an interval on a residue sequence) and a Literal Sequence (a character string of IUPAC character codes[27]), the newer Repeated Sequence enables composition of Alleles that may have clinical significance when represented as a repeating subsequence, such as measuring CAG sequence repeats in Huntington's Disease.[28]

In addition to Literal Sequence and Repeated Sequence Expressions, Alleles may also be composed from Derived Sequence Expressions. A Derived Sequence Expression describes a sequence derived from a location; they are often meant to be taken as "near-equivalent" to the sequence represented within the bounds of the Location's interval. An example of this is describing a tandem duplication of a several kilobase region; rather than specifying the entire precise sequence (including SNVs) that was duplicated, two copies of the region (considered to be approximately/functionally reference matched) may be compactly specified through use of Derived Sequence Expression. The flexibility of these classes allows for the semantically precise representation of many common forms of variation.

### Machine-readable schema

To be useful for information exchange, the information model must be realized in a structured message syntax. VRS specifies its message syntax through a schema currently implemented in JSON Schema. However, the VRS information model could be readily translated to other schema frameworks (e.g., DTD, XSD, Apache Thrift) as desired. For example, the VRS information model has been implemented as Google Protocol Buffers (protobuf) message structures for use in protobuf data streams (implementation at https://github.com/ga4gh/vrs-protobuf). Using the JSON Schema representation enables data consumers to validate VRS objects that are passed between systems. The schema defines the attributes and associated value types for valid VRS objects, and it also includes regular expression validation for relevant attributes (e.g., compact URIs and cytoband descriptions). The VRS repository includes language-agnostic tests for ensuring schema compliance in downstream implementations.

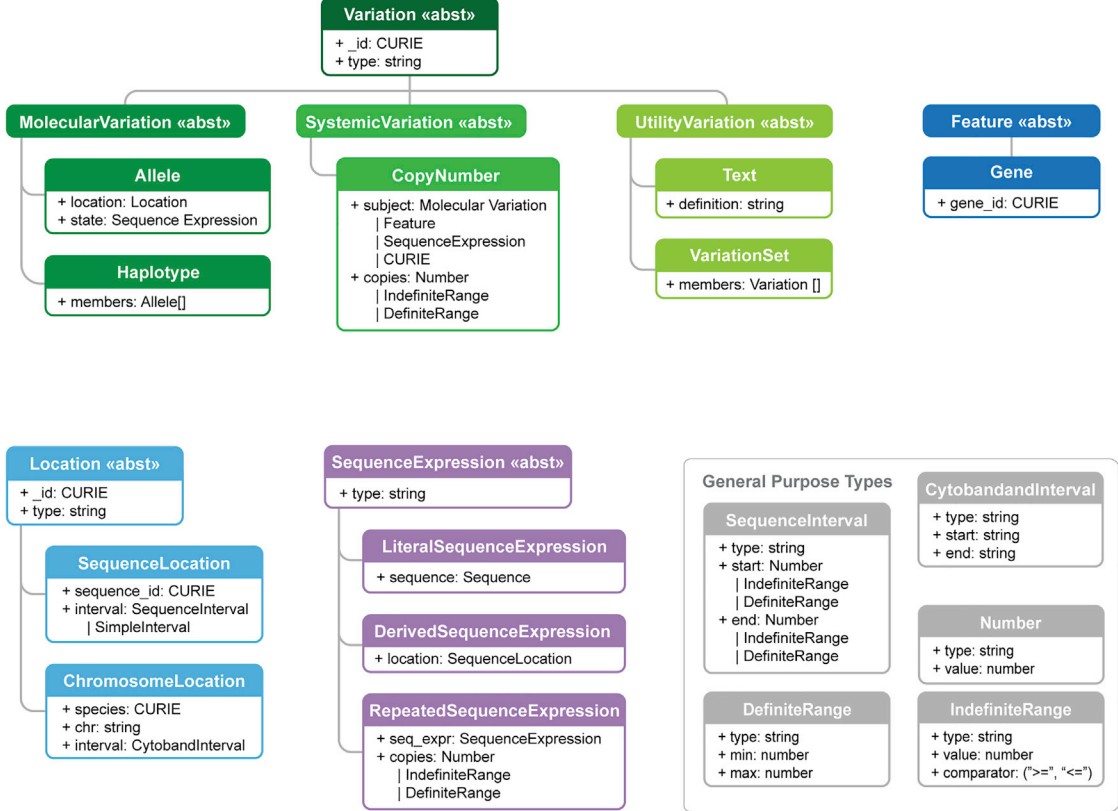

**Figure 2. Information model**

The VRS information model consists of several interdependent data classes, including both concrete classes and abstract superclasses (indicated by <<abst>> stereotype in header). These classes may be broadly categorized as conceptual representations of Variation (green boxes), Feature (blue boxes), Location (light blue boxes), Sequence Expression (purple boxes), and General Purpose Types (gray boxes). The general purpose types support the primary classes, including intervals, ranges, Number, and GA4GH Sequence strings (not shown). While all VRS objects are value objects, only some objects are intended to be identifiable (Variation, Location, and Sequence). Conceptual inheritance relationships between classes is indicated by connecting lines.

### Conventions that promote reliable data sharing

Building upon the terminology, information model, and schema, VRS also provides recommended and required conventions regarding the generation of VRS objects to best facilitate data sharing. While the schema provides the structure of messages, these conventions assist in the evaluation and selection of values to use in VRS objects.

VRS uses inter-residue coordinates for specifying Sequence Locations because they enable a degree of conceptual clarity that is not possible with residue-based coordinate systems. Specifically, residue-counted coordinates require that coordinates are considered exclusively for insertions, but inclusively for substitutions and deletions (Figure 3A). As a consequence, the use of residue coordinates requires knowledge of the operation type (insertion/deletion/substitution) in order to interpret residue-counted coordinates. By choosing an inter-residue coordinate system, VRS is able to construct Location objects that have a singular, immutable interpretation regardless of the variation context.

The term "inter-residue coordinates" is often conflated with other terms, such as "0-based coordinates" or "half-open coordinates." The choice of what to count—residue positions versus inter-residue positions—is critical to decoupling the interpretation of coordinates from the variant type. The choice of whether a number system starts at zero or one is not significant, and description of coordinates as "0-based" risks potential confusion with a hypothetical system where coordinates refer to residues but arbitrarily begins sequences at coordinate 0 instead of the common convention of 1. Therefore, while inter-residue coordinates are numerically equivalent to other coordinate systems for some positions, we emphasize that VRS is strictly based on inter-residue coordinates, and we encourage practitioners to adopt this term and concept to improve the fidelity of data sharing.

There are many sources of ambiguity when representing sequence variants. VRS eliminates some sources of ambiguity and minimizes those that result from conflicting requirements. For example, while VRS eliminates ambiguity that results from human preferences for assigned (rather than computed) identifiers, it cannot completely eliminate ambiguity that results from the conflicting requirements to be able to represent indel variation and sequence repeats as first-class concepts. While semantic distinctions between these concepts exist, the same empirical resulting sequence could be represented with multiple variation expressions.

**A** Residue and Inter-Residue Coordinate System

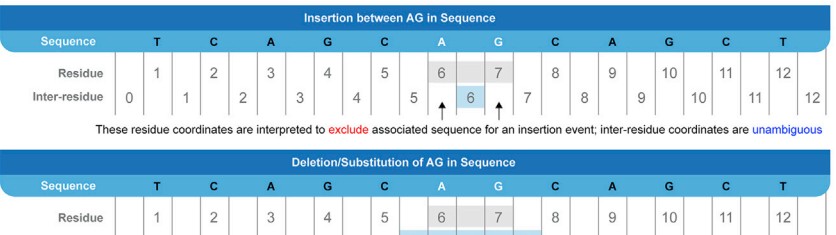

These residue coordinates are interpreted to exclude associated sequence for an insertion event; inter-residue coordinates are unambiguous

The same residue coordinates are interpreted to include associated sequence for a deletion or substitution event; inter-residue coordinates remain unambiguous

**B** Example: An insertion of **GCA** between **AG** in TCAGC**AG**CAGCT

| System | Strategy | Description | Altered Sequence |
|---|---|---|---|
| Actual Event | N/A | Insert GCA between residues 6 and 7 | T C A G C **A G C A G** C A G C T |
| HGVS | 3' rule | 11_12insAGC  or  3AGC[4] | T C A G C **A G** C A G C A G C T |
| VCF | Left-shift | 1-T-TCAG | T C A G C A G C A G C A G C T |
| VRS | Full-justify | {start:1, end:11, state:CAGCAGCAGCAGC} | T C A G C A G C A G C A G C T |

‾‾‾‾ = region of ambiguity

**C** VRS Allele Normalization Algorithm

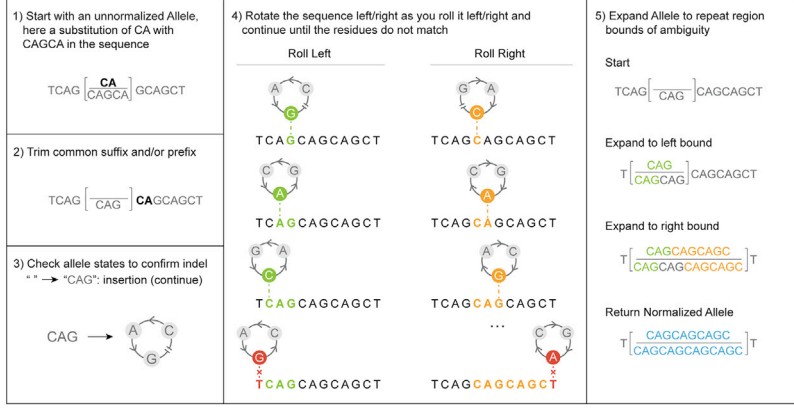

**Figure 3. VRS conventions**
VRS provides many conventions to precisely describe and normalize molecular variation.

(A) A key difference between VRS and other genomic variant formats such as HGVS[20] and VCF[23] is the use of inter-residue coordinates. In this example, the same residue coordinates (gray shading) are used to ambiguously describe the space between two nucleotides where an insertion occurs (top), or the space including the two nucleotides for deletions and substitutions (bottom). Inter-residue coordinates (blue shading) allow for precise representation of nucleotide or inter-nucleotide position without requiring knowledge of the operation, decoupling location representation from the representation of variation.

(B) Here, a three-nucleotide insertion (GCA) occurs in a repetitive region, creating ambiguity as to where the true event (first row) actually occurred. Three systems for describing this variant are depicted. In HGVS (second row), the 3'-most position is selected to represent the insertion. An alternative HGVS representation has the 3'-most position define the repeat unit (here, "AGC"), then the variation is described by the number of repeated units from the first nucleotide of the 5'-most unit in the reference sequence. In VCF (third row), the leftmost insertion point is selected and an "anchor base" prepended to describe the insertion. In contrast to these other systems, VRS (fourth row) avoids the selection of an arbitrary over-precise representation and instead uses a full-justification representation that covers the entire region of ambiguity.

(C) Full-justification Allele normalization is enabled by a specified normalization algorithm. In this example, the unnormalized Allele "reference" and "alternate" sequences (step 0) are trimmed of their common suffix "CA" (step 1). Only the resulting "reference" sequence is blank, indicating this is an insertion, and the algorithm continues (step 2). The non-blank "alternate" sequence is incrementally rolled left to identify the left bound of matching repetitive sequence, then incrementally rolled right to identify the right bound (step 3). These boundaries are used to prepend (step 4, green sequence) and append (step 4, orange sequence) the regions of ambiguity to both sequences, resulting in a normalized, fully justified Allele (step 4, blue sequence).

Normalization is the process by which a representation is converted into a canonical form,[11,29–31] and is an important component of minimizing ambiguity. VRS adopts the use of a fully justified representation (Figure 3B) as a strongly recommended convention, ensuring that insertions and deletions in repetitive regions are not arbitrarily located to a specific position within a sequence but instead describe the alteration over the entire region of ambiguity. This is achieved through VRS Allele Normalization (Figure 3C; STAR Methods), which is an implementation of NCBI's Variant Overprecision Correction Algorithm (VOCA).[29] Notably, the normalization method used to describe ambiguous indels in repetitive regions only applies to Literal Sequence Expression Alleles. Both VRS and HGVS also provide alternate representations to define specific repeated subsequences in these regions; additional normalization rules for normalizing repeated sequence expressions in VRS are also specified. While we strongly recommend that VRS Allele Normalization (and future normalization rules for forthcoming VRS objects) be used in most

implementations, we recognize that there are some rare situations where the structure of VRS objects is sufficient but the general rules for normalization should not apply for variation representation.

### Globally unique computed identifiers

VRS provides an algorithmic solution to deterministically generate a globally unique identifier from a VRS object. All valid implementations of the VRS Computed Identifier will generate the same identifier when the objects are identical and will generate different identifiers when they are not. The VRS Computed Identifier scheme works for all subclasses of Variation in VRS and is intended to be used by other GA4GH specifications, such as the Refget API Specification.[32] In addition to Variation subclasses, Location subclasses are also identifiable and are therefore referenceable by a global computed identifier, though other VRS classes are not meaningful in isolation and are therefore not identifiable.

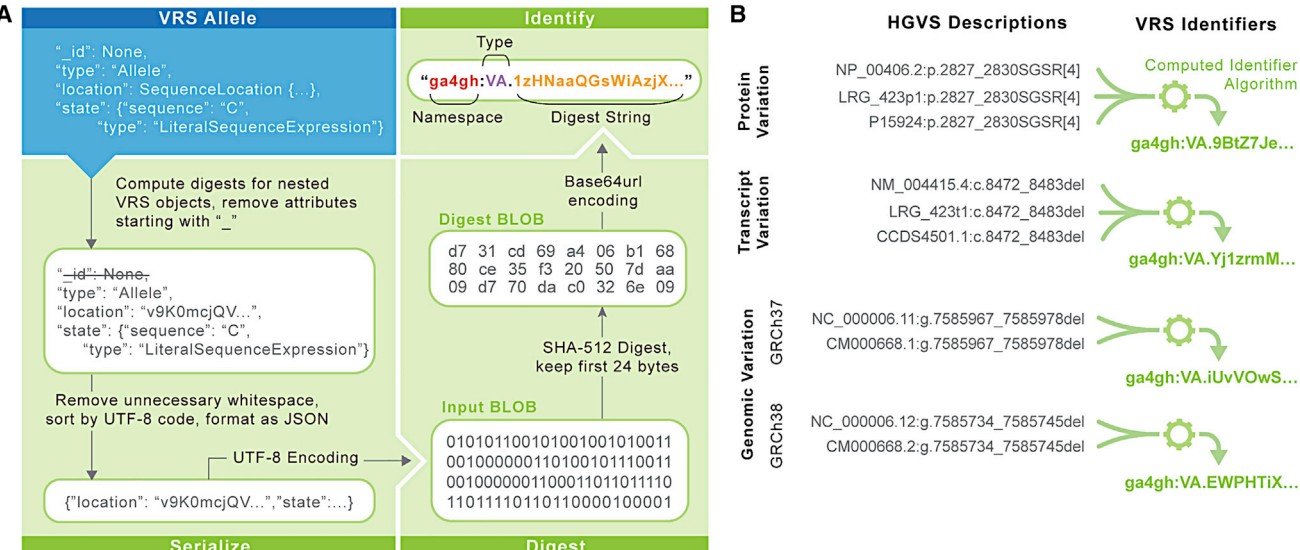

**Figure 4. Computed identifiers**

VRS provides a mechanism for federated variation identification via the Computed Identifier Algorithm.

(A) The Computed Identifier Algorithm is defined in three stages. First, an identifiable VRS object such as an Allele (blue box) is transformed into a well-defined and canonical serialized JSON representation. The serialized Binary Large Object (BLOB) is then digested via the SHA-512 algorithm, truncated to retain only the first 24 bytes, and subsequently encoded using base64url. The resulting digest string (green text) is then appended to the object type identifier; for an Allele object, the identifier prefix is "VA" (blue text). The identifier is then assembled into a compact URI (CURIE) under the ga4gh namespace (orange text).

(B) Use of the VRS framework enables de-duplication of identical variation concepts with differing HGVS descriptions. Here, multiple synonymous HGVS descriptions are indicated for a variant on genome builds GRCh37 and GRCh38, the corresponding transcript variant, and predicted protein translation. These four contexts (two genome assemblies, transcript, and protein) resolve to four distinct identifiers, regardless of which synonymous description is used to build the VRS object. Ellipses ("…") used in objects and strings in this diagram represent content that is omitted for simplicity of presentation. The VRS-Python implementation provides full support for all operations depicted here, including translating between HGVS and VRS formats. For additional details, see https://vrs.ga4gh.org/en/1.2/impl-guide/computed_identifiers.html.

The algorithm for constructing VRS identifiers consists of four operations (Figure 4A; STAR Methods). First, a VRS object is normalized using type-specific rules defined in VRS; normalization applies, in principle, to all object types in order to standardize representation. Second, the object is serialized (i.e., converted into a string) using well-defined rules in VRS, based on a canonical JSON format. Third, a digest of the serialized data is created using the common SHA512 hashing algorithm and truncating the output to 24 bytes, which are subsequently encoded using the IETF standard base64url character set. Finally, the computed digests are prepended by a class-specific identifier prefix (Table 1) and the ga4gh namespace prefix. Identifier prefixes are intended to persist with the underlying data model and terminology, providing transparency into the class of object being described and analogous to the use of ENST prefixes for Ensembl/GENCODE transcript identifiers. Using a 24-byte digest virtually guarantees the uniqueness of variation identifiers; for example, in a hypothetical collection of $10^{18}$ objects, the probability of a single collision is less than $10^{-21}$.[33]

VRS variation identifiers depend on the attributes comprising the variation object, including the reference sequence identity. This is a contrast with variation formats that instead rely upon human-assigned accessions. As a result, semantically equivalent variation objects that are defined on synonymous accessions will result in the same VRS identifier (Figure 4B). VRS computed identifiers are amenable to arbitrary reference sequences, including proprietary sequences or segments of a graph genome. The VRS computed identifier algorithm obviates pre-negotiation of variation identifier namespaces by allowing computational pipelines to generate reliable private identifiers efficiently and on-demand. This lowers barriers for distributed groups to share and collate annotations and interpretations with computed identifiers as keys.

### Implementations and community adoption

We provide a Python package (VRS-Python[34]) that implements the above schema and algorithms and supports translation of VRS to and from the HGVS and Sequence, Position, Deletion, Insertion (SPDI)[29] variant representation schemes to facilitate rapid adoption for genomic data sharing. Support for VCF translation will be addressed in a future version of VRS-Python. VRS has been adopted by, and is being evaluated by, several major institutions in the bioinformatics community, including the ClinGen Allele Registry, the NCBI, the BRCA Exchange, and the VICC MetaKB. Both VRS (https://github.com/ga4gh/vrs) and VRS-Python (https://github.com/ga4gh/vrs-python) are publicly available on GitHub and maintained under the Apache v2.0 license. While VRS-Python may be used as the basis for development in Python, it is not required in order to use VRS. Additional community implementations of VRS have been created in C++ (Table S2). Community feedback, requests, and contributions to these repositories are welcome and

encouraged. Documentation is automatically generated from the VRS repository and made available online for reference at https://vrs.ga4gh.org/.

## DISCUSSION

The Variation Representation Specification is a GA4GH-approved standard, developed through the coordinated effort of variation representation experts and major genomic data providers from industry, government, and academic sectors. It is intended to support exchange of genomic variation data between computational systems, with a focus on semantic precision, extensibility, and conventions to promote reliable, federated identification and search. VRS, operating under the aegis of GA4GH, is well-positioned to collect and adapt to the needs of its users through well-advertised and open forums, including regular working group meetings, GA4GH conferences, mailing lists, GitHub, and Slack.

At its foundation, VRS provides a terminology and information model that are independent of technical choices of messaging protocols or implementation language. Use of these components and the decision to design VRS objects as value objects are a novel approach to variation representation that provides data consumers the necessary tools to reliably send and reconstruct the immutable and precise semantic meaning of a given variant. Importantly, this is done using only the minimal information content provided within VRS objects. Adherence to semantic versioning gives data consumers confidence that VRS objects will be semantically consistent and fully compatible with any future v1.x extensions of the specification. On release of a major version of VRS, tooling will also be released to translate previous major versions of VRS objects to the new major version. In addition, the terminology is freely available, open-source, and permissively licensed, simplifying the use and reuse of VRS objects.

Beyond the information model, VRS provides a schema and implementation guidance to promote variation messages that are consistent across implementations. A language-agnostic test suite and open-source VRS-Python reference implementation are key tools freely available to the community to reduce barriers to entry. In addition, the VRS Computed Identifier Algorithm enables a federated network for the exchange of genomic variation, through the generation of unique computed identifiers. The algorithm is supported by the VRS information model and normalization conventions, and allows genomic data providers to consistently identify variation without prior negotiation between resources. This also enables free exchange of variation data as part of a federated network and reduces the normalization burden on downstream data consumers. Together, these features of VRS provide a framework for interoperability, reinforced by the growing network of services that have implemented VRS-compliant API endpoints.

### Limitations of the study

While the design choices leading to the precise, computable representation of variation through VRS provide new opportunities for genomic data sharing, these decisions also bring new considerations for genomic data providers and leave additional chal-

lenges to be addressed. Labels that are used to describe sequences (e.g., "NC_000001.11," "chr 1," "NM_001374258.1"), a single variant (e.g., "SCV000504256"), or sets of variation (e.g., "VCV000013961.13," "CA123643," "rs113488022," "deltaF508") are extraneous to the minimal information used to construct VRS objects, and so resources wishing to provide descriptors for these concepts must transmit them in parallel with VRS objects or reference endpoints where these descriptions may be retrieved. In practice, GA4GH Driver Projects using VRS have found it desirable to use VRS objects to precisely represent variation and wrap VRS objects inside "Value Object Descriptors" to provide the various human-readable labels describing VRS objects.

A related challenge is that many genomic variation registries aggregate several related variant contexts under a single identifier. Reuse of these resource-provided identifiers then requires downstream data consumers to tease apart the intent of the identifier from the aggregate contexts, a non-trivial exercise. This challenge is compounded when trying to integrate information from multiple sources that make different choices for constructing sets of variant contexts.

The GA4GH GKS Work Stream is working to build complementary standards to VRS to address these challenges. In close collaboration with GA4GH Driver Projects, we are developing a policy for the selection and description of the originating context from aggregate variation identifiers, which is analogous to other community policies such as the Matched Annotation from NCBI and EMBL-EBI (MANE) Select transcript set (https://www.ncbi. nlm.nih.gov/refseq/MANE/). We are also developing a formal specification for value object descriptors, in close coordination with the emerging GA4GH Variation Annotation specification. GKS is also investigating policies and frameworks by which VRS objects can be attached and referenced in written works, to enable the precision and extensibility of computational variation representation with VRS to accompany free-form descriptions in natural language. These and related policies and tools surrounding the use of VRS are being assembled in the VRSATILE framework, available at https://vrsatile.readthedocs.io/.

### Conclusions

As a specification and framework for the federated exchange of genomic variation data, the GA4GH Variation Representation Specification is precise, reproducible, and extensible to all forms of biomolecular variation. It separates the distinct concerns of sequence location and state, and it clarifies the distinctions between molecular and systemic forms of variation. It also provides an integrative collection of components for the description, representation, and validation of variation concepts between systems. Finally, it is accompanied by multiple implementations from major genomic data providers, including an open-source and freely available reference Python implementation. The latest version of the specification is freely available for reference online at https://vrs.ga4gh.org/.

## STAR★METHODS

Detailed methods are provided in the online version of this paper and include the following:

## SUPPLEMENTAL INFORMATION

## ACKNOWLEDGMENTS

The authors thank Christopher Bizon (Renaissance Computing Institute), Karen Eilbeck (University of Utah), Cristina Y. Gonzalez, Tim Hefferon (NCBI), Brad Holmes (NCBI), Anna Lu (National Cancer Institute), Donna R. Maglott (NCBI), Christa Lese Martin (Geisinger), and Lon Phan (NCBI) for important discussions and critical feedback that substantially advanced this work. The authors also thank Ewan Birney (GA4GH, European Molecular Biology Laboratory, European Bioinformatics Institute), Peter Goodhand (GA4GH), and Angela Page (GA4GH) for providing organizational support that enabled this work. A.H.W. was supported by [K99HG010157], [R00HG010157], [R35HG011949] and [U24CA237719]. L.B. and H.L.R. were supported by [U41HG006834] and [U24HG011025]. M.B. was supported by the BioMedIT Network project of SIB & SPHN. M.C. was supported by [U01CA242954]. M.G. was supported by [R00HG007940]. O.L.G. was supported by [U24CA237719]. S.E.H., H.S., K.T., and A.D.Y. were supported by the Wellcome Trust [WT108749/Z/15/Z] and [WT201535/Z/16/Z] and the European Molecular Biology Laboratory. E.M. was supported by the Intramural Research Program of the National Institutes of Health, National Library of Medicine. R.Y.P. and K.R. were supported by [U41HG009650]. R.R.F. was supported by [U41HG006834] and the Mayo Clinic Center for Individualized Medicine. R.K.H. was supported by ClinGen [U41HG006834], Invitae, and MyOme. This research was funded in part by the Wellcome Trust [WT108749/Z/15/Z] and [WT201535/Z/16/Z]. For the purpose of open access, the author has applied a CC BY public copyright license to any Author Accepted Manuscript version arising from this submission.

## AUTHOR CONTRIBUTIONS

Author contributions follow the Contributor Roles Taxonomy (CRediT) conventions. A.H.W., L.B., G.A., M. Baudis, M. Brush, D.L.C., M.C., O.L.G., S.E.H., D.K., J.M.L., J.L., E.M., T.N., R.Y.P., K.R., S.R., H.S., K.T., A.D.Y., R.R.F., and R.K.H. contributed to conceptualization. A.H.W., L.B., G.A., D.L.C., J.M.L., T.N., S.R., K.T., R.R.F., and R.K.H. contributed to methodology. A.H.W., L.B., M.C., E.M., T.N., R.Y.P., K.R., S.R., B.W., and R.K.H. contributed to software. R.Y.P., K.R., B.W., and R.K.H. contributed to validation. A.H.W., L.B., D.L.C., and R.K.H. contributed to formal analysis. A.H.W., L.B., G.A.,

E.M., T.N., R.R.F., and R.K.H. contributed to investigation. A.H.W., L.B., G.A., M. Baudis, M.C., R.Y.P., and K.R. contributed to resources. A.H.W., L.B., M. Baudis, and R.K.H. contributed to data curation. A.H.W. and R.K.H. contributed to writing - original draft. A.H.W., L.B., G.A., M. Baudis, M. Brush, M.C., M.G., O.L.G., S.E.H., D.K., J.M.L., S.L., J.L., E.M., T.N., K.R., P.N.R., H.S., K.T., M.K., H.L.R., A.D.Y., R.R.F., and R.K.H. contributed to writing - review & editing. A.H.W., L.B., S.L., R.R.F., and R.K.H. contributed to visualization. L.B., M. Baudis, M.G., O.L.G., M.K., H.L.R., A.D.Y., R.R.F., and R.K.H. contributed to supervision. A.H.W., L.B., M. Baudis, M.K., H.L.R., A.D.Y., R.R.F., and R.K.H. contributed to project administration. A.H.W., L.B., O.L.G., H.L.R., A.D.Y., R.R.F., and R.K.H. contributed to funding acquisition.

## DECLARATION OF INTERESTS

H.L.R. is a member of the advisory board for *Cell Genomics*.

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

## CellPress

## Cell Genomics
Technology

# STAR★METHODS

## KEY RESOURCES TABLE

| REAGENT or RESOURCE | SOURCE | IDENTIFIER |
|---|---|---|
| Software and algorithms | | |
| Variation Representation Specification Repository | This paper &GitHub repository | https://doi.org/10.5281/zenodo.5518099 |
| VRS-Python Repository | This paper &GitHub repository | https://doi.org/10.5281/zenodo.5525375 |

## RESOURCE AVAILABILITY

### Lead contact
Requests for further information should be directed to and will be fulfilled by the lead contact, Alex Wagner (Alex.Wagner@nationwidechildrens.org).

### Materials availability
This study did not involve the generation of unique reagents.

### Data and code availability

- This study did not involve the generation of biological or chemical data.
- The specification described by this study is available online at https://vrs.ga4gh.org, which is generated from the specification codebase at https://github.com/ga4gh/vrs. The v1.2.0 specification has been deposited at Zenodo, https://doi.org/10.5281/zenodo.5518099. The associated vrs-python utility described by this study has also been deposited at Zenodo, https://doi.org/10.5281/zenodo.5525375.
- Any additional information required to reanalyze the data reported in this paper is available from the lead contact upon request.

## METHOD DETAILS

### The VRS Development Process
The release cycle is implemented in the VR project board, which is the authoritative source of information about development status.

### Planned Features
Feature requests from the community are made through the generation of GitHub issues on the VRS repository, which are open for public review and discussion.

### Project Leadership Review
Open issues are reviewed and triaged by Project Leadership. Feature requests identified to support an unmet need are added to the Backburner project column and scheduled for discussion in our weekly VR calls. These discussions are used to inform whether or not a feature will be planned for development. The Project Maintainers are responsible for making the final determination on whether or not a feature should be added to VRS.

### Requirements Gathering
Once a planned feature is introduced in call, the issue moves to the Planning project column. During this phase, community feedback on use cases and technical requirements will be collected (see example requirement issues). Deadlines for submitting cases will be set by the Project Maintainers.

### Requirements Discussion
Once the requirements gathering phase has been completed, the issue moves to the Backlog/Ready for Dev project column. In this phase, the requirements undergo review and discussion by the community on VR calls.

### Feature Development
After community review of requirements, the issue moves on to the In Progress project column. In this stage, the draft features will be developed as a draft Pull Request (PR). The draft author will indicate that a feature is ready for community review by marking the PR as ''Ready for review'' (at which point the PR loses ''draft'' status).

### Feature Review
Once a PR is ready for review, the Project Maintainers will move the corresponding issue to the QA/Feedback project column. Pull requests ready for public review MAY be merged into the main (stable release) branch through review and approval by at least one

(non-authoring) Project Maintainer. Merged commits MAY be tagged as alpha releases when needed. After merging, corresponding issues are moved to the <u>Done</u> project column and are closed.

## Version Review and Release

After completion of all planned features for a new minor or major version, a request for community review will be indicated by a beta release of the new version. Community stakeholders involved in the feature requests and requirements gathering for the included features are notified by Project Maintainers for review and approval of the release. After a community review period of at least one week, the Project Leadership will review and address any raised concerns for the reviewed version.

After passing review, new minor versions are released to production. If any features in the reviewed version are deemed to be significant additions to the specification by the Project Leadership, or if it is a major version change, instead a release candidate version will be released and submitted for GA4GH product approval. After approval, the new version is released to production.

VRS follows GA4GH project versioning recommendations, based on <u>Semantic Versioning 2.0</u>.

## Leadership
### Project Leadership

As a product of the Genomic Knowledge Standards (GKS) Work Stream, project leadership is comprised of the Work Stream leadership:

- Alex Wagner (<u>@ahwagner</u>; corresponding author)
- Andy Yates (<u>@andrewyatz</u>)
- Bob Freimuth (<u>@rrfreimuth</u>)
- Javier Lopez (<u>@javild</u>)
- Larry Babb (<u>@larrybabb</u>; corresponding author)
- Matt Brush (<u>@mbrush</u>)
- Reece Hart (<u>@reece</u>; corresponding author)

### Project Maintainers

Project maintainers are the leads of the GKS Variation Representation working group:

- Alex Wagner (<u>@ahwagner</u>; corresponding author)
- Larry Babb (<u>@larrybabb</u>; corresponding author)
- Reece Hart (<u>@reece</u>; corresponding author)

## VRS Allele Normalization Algorithm

The VRS Allele Normalization Algorithm is a procedure by which all Alleles–reference-matched, substitutions, insertions, and deletions–are normalized to a precise, unambiguous form. This algorithm was designed for Allele instances in which the *Reference Allele Sequence* and *Alternate Allele Sequence* are precisely known and intended to be normalized. In some instances, this may not be desired, e.g., faithfully maintaining a sequence represented as a repeating subsequence through a *RepeatSequence* object. We also anticipate that these edge cases will not be common, and encourage adopters to use the VRS Allele Normalization Algorithm whenever possible.

The VRS Normalization Algorithm is defined as follows:

0. Start with an unnormalized Allele, with corresponding "reference" and "alternate" Allele Sequences.
    a The *Reference Allele Sequence* refers to the subsequence at the Allele SequenceLocation.
    b The *Alternate Allele Sequence* refers to the Sequence described by the Allele state attribute.
    c Let *start* and *end* initially be the *start* and *end* of the Allele SequenceLocation.
1. Trim common flanking sequence from Allele sequences.
    a Trim common suffix sequence (if any) from both of the Allele Sequences and decrement *end* by the length of the trimmed suffix.
    b Trim common prefix sequence (if any) from both of the Allele Sequences and increment *start* by the length of the trimmed prefix.
2. Compare the two Allele sequences, if:
    a both are empty, the input Allele is a reference Allele. Return the input Allele unmodified.
    b both are non-empty, the input Allele has been normalized to a substitution. Return a new Allele with the modified *start*, *end*, and *Alternate Allele Sequence.*
    c one is empty, the input Allele is an insertion (empty *reference sequence*) or a deletion (empty *alternate sequence*). Continue to step 3.
3 Determine bounds of ambiguity.
    a Left roll: Set a *left_roll_bound* equal to *start.* While the terminal base of the non-empty Allele sequence is equal to the base preceding the *left_roll_bound*, decrement *left_roll_bound* and circularly permute the Allele sequence by removing the last character of the Allele sequence, then prepending the character to the resulting Allele sequence.

b  Right roll: Set a *right_roll_bound* equal to *start.* While the terminal base of the non-empty Allele sequence is equal to the base following the *right_roll_bound*, increment *right_roll_bound* and circularly permute the Allele sequence by removing the first character of the Allele sequence, then appending the character to the resulting Allele sequence.

4  Construct a new Allele covering the entire region of ambiguity.

a  Prepend characters from *left_roll_bound* to *start* to both Allele Sequences.

b  Append characters from *start* to *right_roll_bound* to both Allele Sequences.

c  Set *start* to *left_roll_bound* and *end* to *right_roll_bound*, and return a new Allele with the modified *start*, *end*, and *Alternate Allele Sequence.*

### Normalizing Alleles that are same as reference

The above normalization procedure returns reference-matched Alleles as expressed (step 2a). We believe that in many cases where a referenced-matched Allele is expressed, the intention is to express precisely the indicated sequence as reference, even if it falls in a repetitive region.

VRS STRONGLY RECOMMENDS that Alleles be normalized when generating computed identifiers. The rationale for recommending, rather than requiring, normalization is grounded in dual views of Allele objects with distinct interpretations:

- **Allele as minimal representation of a change in sequence.** In this view, normalization is a process that makes the representation minimal and unambiguous.
- **Allele as an assertion of state.** In this view, it is reasonable to want to assert state that may include (or be composed entirely of) reference bases, for which the normalization process would alter the intent.

Although this rationale applies only to Alleles, it may have have parallels with other VRS types. In addition, it is desirable for all VRS types to be treated similarly.

Furthermore, if normalization were required in order to generate Computed Identifiers, but did not apply to certain instances of VRS Variation, implementations would likely require secondary identifier mechanisms, which would undermine the intent of a global computed identifier.

The primary downside of not requiring normalization is that Variation objects might be written in non-canonical forms, thereby creating unintended degeneracy.

Therefore, normalization of all VRS Variation classes is optional in order to support the view of Allele as an assertion of state on a sequence.

### VRS Serialization Procedure

Digest serialization converts a VRS object into a binary representation in preparation for computing a digest of the object. The Digest Serialization specification ensures that all implementations serialize variation objects identically, and therefore that the digests will also be identical. VRS provides validation tests to ensure compliance.

Although several proposals exist for serializing arbitrary data in a consistent manner, none have been ratified. The VRS Serialization Procedure for creating computed identifiers is distinct from JSON serialization or other serialization forms. Although Digest Serialization and JSON serialization appear similar, they are NOT interchangeable and will generate different GA4GH Digests. As a result, VRS defines a custom serialization format that is consistent with these proposals but does not rely on them for definition; it is hoped that a future ratified standard will be forward compatible with the process described here.

The first step in serialization is to generate message content. If the object is a string representing a Sequence, the serialization is the UTF-8 encoding of the string. Because this is a common operation, implementations are strongly encouraged to precompute GA4GH sequence identifiers as described in Required External Data.

- If the object is a composite VRS object, implementations MUST:
- ensure that objects are referenced with identifiers in the ga4gh namespace
- replace nested identifiable objects (i.e., objects that have id properties) with their corresponding digests
- order arrays of digests and ids by Unicode Character Set values
- filter out fields that start with underscore (e.g., _id)
- filter out fields with null values

The second step is to JSON serialize the message content with the following REQUIRED constraints:

- encode the serialization in UTF-8
- exclude insignificant whitespace, as defined in RFC8259§2
- order all keys by Unicode Character Set values
- use two-char escape codes when available, as defined in RFC8259§7

The criteria for the digest serialization method was that it must be relatively easy and reliable to implement in any common computer language.

### Required External Data

All VRS implementations will require external data regarding sequences and sequence metadata. The choices of data sources and access methods are left to implementations. This section provides guidance about how to implement required data and helps implementers estimate effort. This section is descriptive only: it is not intended to impose requirements on interface to, or sources of, external data. For clarity and completeness, this section also describes the contexts in which external data are used:

**Conversion from other variant formats.** When converting from other variation formats, implementations MUST translate primary database accessions or identifiers (e.g., *NM_000551.3* or *refseq:NM_000551.3*) to a GA4GH VRS sequence identifier (*ga4gh:SQ.v_QTc1p-MUYdgrRv4LMT6ByXIOsdw3C_*)

**Conversion to other variant formats.** When converting to other variation formats, implementations SHOULD translate GA4GH VR sequence identifier (*ga4gh:SQ.v_QTc1p-MUYdgrRv4LMT6ByXIOsdw3C_*) to primary database identifiers (*refseq:NM_000551.3*) that will be more readily recognized by users.

**Normalization.** During normalization, implementations will need access to sequence length and sequence contexts.

The data required in the above contexts is summarized in the Data Service Descriptions table:

### Data Service Descriptions

| Data Service | Description | Contexts |
|---|---|---|
| sequence | For a given sequence identifier and range, return the corresponding subsequence. | normalization |
| sequence length | For a given sequence identifier, return the length of the sequence | normalization |
| identifier translation | For a given sequence identifier and target namespace, return all identifiers in the target namespace that are equivalent to the given identifier. | Conversion to/from other formats |

In order to maximize portability and to insulate implementations from decisions about external data sources, implementers should consider writing an abstract data proxy interface to define a service, and then implement this interface for each data backend to be supported. The data proxy interface defines three methods:

- *get_sequence(identifier, start, end)*: Given a sequence identifier and start and end coordinates, return the corresponding sequence segment.
- *get_metadata(identifier)*: Given a sequence identifier, return a dictionary of length, alphabet, and known aliases.
- *translate_sequence_identifier(identifier, namespace)*: Given a sequence identifier, return all aliases in the specified namespace. Zero or more aliases may be returned.

The vrs-python Data Proxy class provides an example of this design pattern and sample replies. GA4GH VRS-Python Implementation implements the DataProxy interface using a local SeqRepo instance backend and using a SeqRepo REST Service backend. A GA4GH refget implementation has been started, but is pending interface changes to support lookup using primary database accessions.

## QUANTIFICATION AND STATISTICAL ANALYSIS

### Hash collision analysis

We define our variables as below:

$P$ = Probability of collision
$b$ = digest length, in bits
$m$ = number of messages in corpus
The length of individual messages is irrelevant.
We calculated the digest length $b$ needed to achieve a collision probability less than $P$ for $m$ messages, using the equation:

$$b(m, P) = \log_2\left(\frac{m^2}{P}\right) - 1$$

Details on the derivation of this equation are described elsewhere.[33]

## ADDITIONAL RESOURCES

VRS Documentation Site: https://vrs.ga4gh.org

