## [Document S2. Transparent peer review records for Wagner et al · Cell Genomics]

The GA4GH Variation Representation Specification (VRS): a Computational Framework for Variation Representation and Federated Identification

Alex H Wagner^{1,2*#}, Lawrence Babb^{3*}, Gil Alterovitz^{4,5}, Michael Baudis⁶, Matthew Brush⁷, Daniel L Cameron^{8,9}, Melissa Cline¹⁰, Malachi Griffith¹¹, Obi L Griffith¹¹, Sarah E Hunt¹², David Kreda¹³, Jennifer M Lee¹⁴, Stephanie Li¹⁵, Javier Lopez¹⁶, Eric Moyer¹⁷, Tristan Nelson¹⁸, Ronak Y Patel¹⁹, Kevin Riehle¹⁹, Peter N Robinson²⁰, Shawn Rynearson²¹, Helen Schuilenburg¹², Kirill Tsukanov¹², Brian Walsh⁷, Melissa Konopko¹⁵, Heidi L Rehm^{3,22}, Andrew D Yates¹², Robert R Freimuth²³, Reece K Hart^{3,24*}

Summary

Scientific Editor:	Orli Bahcall
Initial submission:	1/21/2021
Revision received:	7/12/2021
Accepted:	9/02/2021
Rounds of review:	2
Number of reviewers:	3

Referee reports, first round of review

Reviewer #1: "The GA4GH Variation Representation Specification (VRS): a Computational Framework for the Precise Representation and Federated Identification of Molecular Variation" by Alex H Wagner, et al.

Authors from a wide range of institutions and specializations present a highly normalized and detailed system for automated interchange of not only sequence variants, but also systemic variation such as copy number variation and transcript expression levels. It is expressly not intended to be "human-readable" in the manner of existing, relatively concise variant representations such as HGVS and SPDI; this frees it to use techniques such as inter-residue coordinates, truncated SHA512 hashes, and nested objects from a class hierarchy to achieve both a higher precision and broader scope of expression.

The manuscript defines the scope of VRS, as well as necessary considerations that fall outside of VRS's scope, and describes other GA4GH efforts such as the Value Object Descriptor framework and Variant Origination Policy. It describes the terminology, information model, machine-readable schema, coordinate system, indel normalization method, and globally unique computed identifiers that enable the precision of VRS. A reference implementation provides an additional resource for adopters, and multiple community implementations have provided the opportunity for the concepts to be tested and refined in practice.

Extensibility is mentioned more than 10 times as a goal and feature of VRS. VRS's information model is based on abstract classes that can be extended to new classes by inheritance, a familiar concept to anyone who has worked with object-oriented programming languages. Semantic versioning is mentioned as ensuring consistence and compatibility with future extensions (more on that later), and Figure 1 shows possible added conventions and alternate implementations as modes of extension. While extensions will be necessary as VRS grows to encompass more forms of variation, I think there is a tension between extensions and interoperability of aging implementations, and extensions must be made with caution. For example, the computed identifier derived from a serialized object will change if the object gains a new field. The new field may add valuable information, but it will come at the cost of incompatible identifiers (unless it is prefixed with an underscore to exempt it from serialization). Similarly, VRS includes an immediate escape hatch from its own specificity in the form of the Text subclass of Variation, which is completely unconstrained in content. The laziest implementation of VRS could simply express any variant (or variation) as Text, although it would not realize any of the benefits of VRS. While such an escape hatch is a necessity when building a real-world system in which VRS cannot yet fully and precisely describe every record of every database that uses it, I think it should come with warning signs and encouragement to use it only as a temporary measure. I don't see any discussion in the manuscript of those tradeoffs between extensibility and flexibility on the one hand, and reliability and interoperability on the other. Perhaps that should be common sense, but is common sense really common?

Versioning is not mentioned before the Conclusions section, in which semantic versioning is asserted to provide "confidence that VRS objects will be semantically consistent and fully compatible with any future v1.x extensions of the specification." It wouldn't hurt to add a few words about why (probably that the meaning of existing fields will not change, although new fields that can be ignored by older implementations may appear in future 1.x versions?). I suppose that assurance is why VRS objects appear from Figure 2 not to have any in-band indication of version (which could be underscore-prefixed and omitted from serialization to avoid breaking identifiers).

Before publication, I would like the authors to clarify a few spots in manuscript:

In "Conventions that promote reliable data sharing", 4th paragraph, the VRS Allele Normalization Algorithm is described as "an extension of NCBI's Variant Overprecision Correction Algorithm", but I don't see the difference. In the supplemental info of [28], VOCA is described in more flowery terms such as "bud" (the minimal, left-shifted representation as in step 2 of Figure 3C and Supplementary Methods) and "blossom" (the expanded allele in step 4 of Figure 3C and Supplementary Methods), but the process and the results of VOCA and the VRS Allele Normalization Algorithm seem the same to me. What is the extension?

The next sentence, "Notably, the normalization method used to describe ambiguous insertions in repetitive regions only applies to Literal Sequence insertions, analogous to the use of the 3' rule for HGVS insertions in these regions" was a stumbling block for me. Is the extension relative to VOCA what we're noting here? (Probably not.) The normalization applies only to Literal Sequence insertions, as opposed to ____? Wouldn't normalization also apply to deletions with ambiguous placement? Wait, is VRS abandoning normalization in some cases and going 3' like HGVS instead? Is that the extension? (But that wouldn't make any sense.) I suspect that the sentence was intended to convey something much less surprising. Was the intention simply to point out that normalization is not applicable to the vast majority of variants because it applies only to pure deletions or pure insertions in repetitive regions (or at least, regions in which the non-empty allele can be rolled at least 1 base)?

In Figure 2, AbsoluteAbundance and RelativeAbundance have thick solid borders indicating that they are Identifiable, which I suppose means that they may be serialized and encoded as IDs. However, Range and the subclasses of SequenceInterval have thin solid borders indicating that, while Concrete, they are not Identifiable. Both Range and Interval are simply a pair of numeric bounds. Why would one be Identifiable but not the other? Or should AbsoluteAbundance and RelativeAbundance have thin borders?

[Searching for "identifiable" in the spec itself seems to lead to a circular definition: "identifiable" is defined as "may be referenced with an identifier" (<https://vrs.ga4gh.org/en/stable/schema.html?highlight=identifiable#overview>) or "have id properties" (https://vrs.ga4gh.org/en/stable/impl-guide/computed_identifiers.html?highlight=identifiable#digest-serialization), while `_id` is "available to identifiable objects" (https://vrs.ga4gh.org/en/stable/terms_and_model.html?highlight=identifiable#optional-attributes). There is also the statement "As value objects, VRS objects are used as primitive types and SHOULD NOT be used

as containers for related data. Instead, related data should be associated with VRS objects through identifiers." (http://vrs.ga4gh.org/en/stable/terms_and_model.html?highlight=identifiable#data-model-notes-and-principles) I see the diagram at <https://vrs.ga4gh.org/en/stable/schema.html?highlight=identifiable#overview> does not yet include AbsoluteAbundance and RelativeAbundance, and has State/SequenceState instead of SequenceExpression and subclasses shown in Figure 2.]

"VRS Serialization Procedure" in Supplemental Methods refers to "Required External Data" which as far as I can tell is not defined in the manuscript -- could provide a link to https://vrs.ga4gh.org/en/stable/impl-guide/required_data.html .

Minor - optional

It seems to me that stronger words than "recommended conventions" should be used for the use of inter-residue coordinates -- the coordinate system is a fundamental property of the specification and critical to the correctness of objects. Should reliable data sharing be promoted by encouraging polite adherence to recommended conventions, or rather ensured by an absolute requirement that a specific coordinate system be used? This is a specification after all. (I am relieved see RFC-style "MUST" and "REQUIRED" in https://vrs.ga4gh.org/en/stable/terms_and_model.html#simpleinterval .)

The term CURIE is used in a couple places without a full definition of the acronym (what is the E? Could refer to <https://www.w3.org/TR/curie/> which I see doesn't explain the E either). Is CURIE in this context a synonym for VRS's globally unique computed identifier? (From the spec, I see it's not necessarily a computed identifier except when serializing.)

The reference given for IUPAC codes [25] suggests adding case differences, bolding and underlining in order to denote relative abundance of nucleotides within ambiguous categories... does VRS really encourage that? If not, then the Cornish-Bowden 1984 recommendations (<https://www.ncbi.nlm.nih.gov/pmc/articles/PMC341218/>, in the references of [25], thanks for leading me to that!) might be a more appropriate reference.

Typo

Conclusions, final paragraph, 3rd sentence seems to be missing a subject (deletion of "It" ?).

Congratulations to the authors on the culmination of a lot of painstaking work. VRS promises to greatly increase the reliability of exchange of not only variants, but also variation information.

Angie Hinrichs

Reviewer #2: This is an important piece of work. It will enable interoperability of genomic data and with that aggregation and systematic analysis. However, the paper needs to be written for an audience whose eminent expertise lies in genomic research, and not in computer science and modelling of data. Therefore, this reviewer believes the paper would benefit majorly by improving the text:

Description of the model

VRS is attempting to provide a precise model for genomic variations, which is currently described using human language. The inclined reader will try attempting to do two things: Create a mental link from how the data are handled today to the VRS, and to understand the choices of model components made. To facilitate either mental task the authors may consider extensive use of examples. For example, the components "LiteralSequence", "DerivedSequence" and "RepeatedSequence" might be intuitively clear, but an example would confirm the intuition and also elucidate how these choices of components came about. It would also help understand the relationship between the components (e.g. is a RepeatedSequence a DerivedSequence or a LiteralSequence, or are they orthogonal to each other?).

The authors assert that "Variation, Location, State, and Interval are all extensible abstract base classes".

However, Figure 2 does not contain a reference to the State class. Instead has an orange SequenceExpression class, which is not mentioned in the text but is perhaps the missing base class.

Figure 3A is difficult to understand. The three examples Insertion, Deletion and Substitution are completely identical, with the exception of coloring inter-residue 6 in the Insertion, and 5, 6 and 7 in the other two. How the two residue notations differ in the resulting description is not described. Figure 3B appears to be an example based on 3A, but the logic of coordinate assignment is not clear.

The description of a "globally unique identifier" lacks a rationale about the advantages of such a concept and why such thing is valuable beyond the canonical JSON. The detailed technical algorithm may be beyond relevance for readers of Cell Genomics. It is also not clear to this reviewer how different sequence identifiers lead to an identical VRS identifier, unless they are resolved to their underlying sequences.

Critique of the model

The authors do not discuss the ambiguity of the model. Unfortunately, one and the same variation can be described in two or more different ways without a mechanism of avoidance or disambiguation, leading to problems with processing by the computer. For example, a RepeatedSequence might easily be represented by a LiteralSequence. A substitution can be modeled as a combination of a deletion and insertion.

Reviewer #3: In their manuscript, "The GA4GH Variation Representation Specification (VRS): A Computational Framework for the Precise Representation and Federated Identification of Molecular Variation", the authors present the key elements of VRS. These include the scope, a terminology/information model, a JSON schema, recommended conventions, globally unique computed identifiers, and tools to facilitate community adoption. The authors emphasize that VRS is intended as a computational framework offering precision, expressiveness, and extensibility in variant representation, rather than (another) human readable variation format. They suggest that VRS complements these other formats.

VRS is distinguished by several features: (1) the use of value objects, (2) distinction for the concepts of molecular and systemic variation, (3) use of inter-residue coordinates for specifying sequence locations, and (4) globally unique computed identifiers (suitable for use by other GA4GH specifications). The primary strength of VRS is in its ability to promote accurate data exchange of variants- because it provides unique identifiers for variant objects, it should be possible to share data about variants regardless of their origin and/or any local or repository-specific identifiers.

It is notable that the authors of this manuscript hail from some of the largest and well-known institutions that generate, store, and evaluate variant data, and have an international distribution. This is meaningful as it suggests the existence of broad support for VRS, improving the likelihood of adoption. Provisions to promote adoptability, such as a translatable schema and packages that translate VRS to existing formats (HGVS and SPDI) should also help in this area. However, it still remains to be seen whether bioinformatics approaches to variant analysis have become sufficiently pervasive to that it will result in this computationally-focused specification becoming the lingua franca for variation.

The manuscript is well-written, as is the accompanying online documentation at <https://vrs.ga4gh.org/en/stable/index.html>.

I support the publication of this manuscript, with no substantial revisions. Several minor items for the authors to address are listed below.

1. Figure 2 Information Model and the schema on the website (<https://vrs.ga4gh.org/en/stable/schema.html>) have some differences. Can those be harmonized?
2. Section on implementation and community adoption: The text mentions that VRS has been adopted by and evaluated by several major institutions in the bioinformatics community (and then lists several by name). Can they more clearly distinguish who has adopted vs. who is evaluating at this time?
3. VRS allele normalization (full justification) requires expanding boundaries to find non-ambiguous boundaries. Within the human genome, there are some regions of multi-megabase length repeats (and with the recent sequencing of human centromeres and telomeres, we can expect to see more variation reported in large repeats). What, if any, impact might these very large normalized allele representations have on storage and performance?
4. If normalization is so strongly encouraged, why isn't it required?

5. The Computed VRS Identifiers provide unique identifiers for variation on the same molecule. Does the model provide any mechanisms for indicating equivalence or non-equivalence of variants on different molecules? (e.g. transcript to genome or genome version 1 to genome version 2)?

Author response to the first round of review

Reviewer/editor text in black, below. Our responses are in red.

From Editor:

I hope this message finds you well. Thank you again for preparing your manuscript on VRS and the opportunity to review your manuscript at Cell Genomics!

We have now received the final reviewer report on your paper, and a copy of all 3 reports are attached below. We invite you to revise your manuscript in response to these comments.

We recommend presenting the manuscript in our Technical Report format, and I would be glad to discuss your preferences. I will also be glad to discuss how to best focus and present this work, in context of this manuscript, and as part of the special issue.

We hope that you will be able to suitably revise within the next 4 weeks. If you will need additional time, please keep us updated on your progress. I will be delighted to discuss these and additional editing suggestions and your plans for revising, to support you in the optimal presentation for this exciting and impactful publication.

Reviewers' Comments:

Reviewer #1:

"The GA4GH Variation Representation Specification (VRS): a Computational Framework for the Precise Representation and Federated Identification of Molecular Variation" by Alex H Wagner, et al.

Authors from a wide range of institutions and specializations present a highly normalized and detailed system for automated interchange of not only sequence variants, but also systemic variation such as copy number variation and transcript expression levels. It is expressly not intended to be "human-readable" in the manner of existing, relatively concise variant representations such as HGVS and SPDI; this frees it to use techniques such as inter-residue coordinates, truncated SHA512 hashes, and nested objects from a class hierarchy to achieve both a higher precision and broader scope of expression.

The manuscript defines the scope of VRS, as well as necessary considerations that fall outside of VRS's scope, and describes other GA4GH efforts such as the Value Object Descriptor framework and Variant Origination Policy. It describes the terminology, information model, machine-readable schema, coordinate system, indel normalization method, and globally unique computed identifiers that enable the precision of VRS. A reference implementation provides an additional resource for adopters, and multiple community implementations have provided the opportunity for the concepts to be tested and refined in practice.

Extensibility is mentioned more than 10 times as a goal and feature of VRS. VRS's information model is based on abstract classes that can be extended to new classes by inheritance, a familiar concept to anyone who has worked with object-oriented programming languages. Semantic versioning is mentioned as ensuring consistence and compatibility with future extensions (more on that later), and Figure 1 shows possible added conventions and alternate implementations as modes of extension. While extensions will be necessary as VRS grows to encompass more forms of variation, I think there is a tension between extensions and interoperability of aging implementations, and extensions must be made with caution. For example, the computed identifier derived from a serialized object will change if the object gains a new field. The new field may add valuable information, but it will come at the cost of incompatible identifiers (unless it is prefixed with an underscore to exempt it from serialization). Similarly, VRS includes an immediate escape hatch from its own specificity in the form of the Text subclass of Variation, which is completely unconstrained in content. The laziest implementation of VRS could simply express any variant (or variation) as Text, although it would not realize any of the benefits of VRS. While such an escape hatch is a necessity when building a real-world system in which VRS cannot yet fully and precisely describe every record of every database that uses it, I think it should come with warning signs and encouragement to use it only as a temporary measure. I don't see any discussion in the manuscript of those tradeoffs between extensibility and flexibility on the one hand, and reliability and interoperability on the other. Perhaps that should be common sense, but is common sense really common?

We have added a sentence to the introductory paragraph of the major header entitled "The Variation Representation Specification" that helps anchor why Value Object representation is important and concretely stating that the objects themselves are not user-extensible. We also added a minor heading entitled "Technical foundations and design decisions" that provides some discussion of how VRS is intended to be extended by addition and reuse of common "building blocks", and added Supplemental Figure 2 to illustrate some current applications of this design.

Versioning is not mentioned before the Conclusions section, in which semantic versioning is asserted to provide "confidence that VRS objects will be semantically consistent and fully compatible with any future v1.x extensions of the specification." It wouldn't hurt to add a few words about why (probably that the meaning of existing fields will not change, although new fields that can be ignored by older implementations may appear in future 1.x versions?). I suppose that assurance is why VRS objects appear from Figure 2 not to have any in-band indication of version (which could be underscore-prefixed and omitted from serialization to avoid breaking identifiers).

We also added a paragraph to the "Technical foundations and design decisions" explicitly describing our versioning practices through the use of the broadly-adopted SemVar standard.

Before publication, I would like the authors to clarify a few spots in manuscript:

In "Conventions that promote reliable data sharing", 4th paragraph, the VRS Allele Normalization Algorithm is described as "an extension of NCBI's Variant Overprecision Correction Algorithm", but I don't see the difference. In the supplemental info of [28], VOCA is described in more flowery terms such as "bud" (the minimal, left-shifted representation as in step 2 of Figure 3C and Supplementary Methods) and "blossom" (the expanded allele in step 4 of Figure 3C and Supplementary Methods), but the process and the results of VOCA and the VRS Allele Normalization Algorithm seem the same to me. What is the extension?

VOCA refers both to an algorithm and an implementation; the implementation was not released. One of the authors of VOCA reviewed the VRS normalization algorithm and stated that it was "essentially the same" as VOCA (Brad Holmes, personal communication), with the exception that VRS normalization operates on reference agreement alleles as a special case. In order to clarify that VRS was not using the VOCA implementation, BH & RKH agreed to use the term "fully-justified" to refer to the outcome of both processes; that is, VOCA and VRS normalization both result in "fully-justified variants". In the manuscript, we have strengthened our language to state that we have *implemented* VOCA and extended our normalization routine to cover reference alleles.

The next sentence, "Notably, the normalization method used to describe ambiguous insertions in repetitive regions only applies to Literal Sequence insertions, analogous to the use of the 3' rule for HGVS insertions in these regions" was a stumbling block for me. Is the extension relative to VOCA what we're noting here? (Probably not.) The normalization applies only to Literal Sequence insertions, as opposed to ____? Wouldn't normalization also apply to deletions with ambiguous placement? Wait, is VRS abandoning normalization in some cases and going 3' like HGVS instead? Is that the extension? (But that wouldn't make any sense.) I suspect that the sentence was intended to convey something much less surprising. Was the intention simply to point out that normalization is not applicable to the vast majority of variants because it applies only to pure deletions or pure insertions in repetitive regions (or at least, regions in which the non-empty allele can be rolled at least 1 base)?

One of the features of VRS is the ability to express multiple forms of variation and the corresponding ambiguity as reported by different types of assays. For substitutions and indels called from NGS data, these are representable as *Literal Sequence Expressions*. For some assays, the reported variants may be represented as copies of a known repeating subsequence. One example of this are the PCR-based electrophoresis assays for reporting CAG repeats in the Huntingtin (HTT) gene. In these cases, Alleles are better represented using *Repeated Sequence Expressions*. In other cases, systems may wish to express insertions or substitutions of large sequences from other places in the genome; in these cases it is appropriate to describe variation using *Derived Sequence Expressions*.

The confusion the reviewer experienced when encountering this text is due to our omission of this version 1.2 feature in our manuscript; beyond this brief mention we did not describe Sequence Expressions in our first submission, and have revised the text and Figure 2 to describe this feature. We also expanded the *Scope of VRS and other GA4GH genomic knowledge standards* section to draw attention to the broader goal of VRS to capture the many forms of variation, and added Supplemental Figure 1 to illustrate the complexity of pathways leading to different forms of variation.

In Figure 2, AbsoluteAbundance and RelativeAbundance have thick solid borders indicating that they are Identifiable, which I suppose means that they may be serialized and encoded as IDs. However, Range and the subclasses of SequenceInterval have thin solid borders indicating that, while Concrete, they are not Identifiable. Both Range and Interval are simply a pair of numeric bounds. Why would one be Identifiable but not the other? Or should AbsoluteAbundance and RelativeAbundance have thin borders?

All forms of Variation (including Abundance concepts such as Copy Number Variation), and all Locations, are identifiable. These represent reusable, contextualized objects that are useful to have global computed identifiers for. Other classes do not convey useful information unless in the context of a containing object, and so they are not identifiable. For example, an Interval is meaningless unless it is

used in the context of a Location; so the Interval object is not identifiable but the Location (which contains an interval) is. We have added text to the subsection with header "Globally unique computed identifiers" that describes this rationale.

[Searching for "identifiable" in the spec itself seems to lead to a circular definition: "identifiable" is defined as "may be referenced with an identifier" (<https://vrs.ga4gh.org/en/stable/schema.html?highlight=identifiable#overview>) or "have id properties" (https://vrs.ga4gh.org/en/stable/impl-guide/computed_identifiers.html?highlight=identifiable#digest-serialization), while `_id` is "available to identifiable objects" (https://vrs.ga4gh.org/en/stable/terms_and_model.html?highlight=identifiable#optional-attributes). There is also the statement "As value objects, VRS objects are used as primitive types and SHOULD NOT be used as containers for related data. Instead, related data should be associated with VRS objects through identifiers." (http://vrs.ga4gh.org/en/stable/terms_and_model.html?highlight=identifiable#data-model-notes-and-principles) I see the diagram at <https://vrs.ga4gh.org/en/stable/schema.html?highlight=identifiable#overview> does not yet include AbsoluteAbundance and RelativeAbundance, and has State/SequenceState instead of SequenceExpression and subclasses shown in Figure 2.]

All of the above is true of identifiable objects; they contain an identifier slot labeled "`_id`" which MAY be used to transmit the identifier. This field is dropped before serialization and digest, as illustrated in manuscript Figure 4A. In practice, this field is utilized in `vrs-python`, but typically stored external to the VRS object when used in larger applications (the pattern of maintaining the identifier externally is by VRSATILE variation descriptors, for example).

The default version of the documentation (with `/stable` in the URL) points to the latest release of VRS. Our current work on version 1.2 was in a "release candidate" state at the time of the original submission. The version 1.2 documentation may be found at <https://vrs.ga4gh.org/en/1.2/>. We have updated this latest version to align with the figures presented in the revised manuscript. This includes some changes to the 1.2 structure for abundance, which has been revised in our most recent release candidate in response to community feedback and discussion.

"VRS Serialization Procedure" in Supplemental Methods refers to "Required External Data" which as far as I can tell is not defined in the manuscript -- could provide a link to https://vrs.ga4gh.org/en/stable/impl-guide/required_data.html.

We made this change.

Minor - optional

It seems to me that stronger words than "recommended conventions" should be used for the use of inter-residue coordinates -- the coordinate system is a fundamental property of the specification and critical to the correctness of objects. Should reliable data sharing be promoted by encouraging polite adherence to recommended conventions, or rather ensured by an absolute requirement that a specific coordinate system be used? This is a specification after all. (I am relieved see RFC-style "MUST" and "REQUIRED" in https://vrs.ga4gh.org/en/stable/terms_and_model.html#simpleinterval.)

We agree that inter-residue coordinates are a requirement of the specification, and have made modifications to the text to clarify this. We are less stringent about requiring our conventions for normalization, due to the way implementations may wish to represent reference-matched Alleles. This was a design decision that reflects a known trade-off, though we fully appreciate the reviewer's perspective. We document our rationale in our "Implementations should normalize" section in the specification (https://vrs.ga4gh.org/en/stable/appendices/design_decisions.html#implementations-should-normalize).

The term CURIE is used in a couple places without a full definition of the acronym (what is the E? Could refer to <https://www.w3.org/TR/curie/> which I see doesn't explain the E either). Is CURIE in this context a synonym for VRS's globally unique computed identifier? (From the spec, I see it's not necessarily a computed identifier except when serializing.)

Thanks for pointing out this oversight. We added text to the specification and the manuscript describing a CURIE. The "E" is meaningless, and used in the acronym to assist in pronunciation.

The reference given for IUPAC codes [25] suggests adding case differences, bolding and underlining in order to denote relative abundance of nucleotides within ambiguous categories... does VRS really encourage that? If not, then the Cornish-Bowden 1984 recommendations (<https://www.ncbi.nlm.nih.gov/pmc/articles/PMC341218/>, in the references of [25], thanks for leading me to that!) might be a more appropriate reference.

Good suggestion, we have made this revision.

Typo

Conclusions, final paragraph, 3rd sentence seems to be missing a subject (deletion of "It" ?).

Fixed, thanks!

Congratulations to the authors on the culmination of a lot of painstaking work. VRS promises to greatly increase the reliability of exchange of not only variants, but also variation information.

Angie Hinrichs

Thank you for the thorough and insightful review Angie!

Reviewer #2:

This is an important piece of work. It will enable interoperability of genomic data and with that aggregation and systematic analysis. However, the paper needs to be written for an audience whose eminent expertise lies in genomic research, and not in computer science and modelling of data. Therefore, this reviewer believes the paper would benefit majorly by improving the text:

We appreciate the reviewer's comments about the importance of this work and agree that the paper should be accessible to a broad audience. We have modified the text as noted below, in response to the areas of feedback.

Description of the model

VRS is attempting to provide a precise model for genomic variations, which is currently described using human language. The inclined reader will try attempting to do two things: Create a mental link from how the data are handled today to the VRS, and to understand the choices of model components made. To facilitate either mental task the authors may consider extensive use of examples. For example, the components "LiteralSequence", "DerivedSequence" and "RepeatedSequence" might be intuitively clear, but an example would confirm the intuition and also elucidate how these choices of components came about. It would also help understand the relationship between the components (e.g. is a RepeatedSequence a DerivedSequence or a LiteralSequence, or are they orthogonal to each other?).

We have added descriptions and examples of the three sequence expression classes to the manuscript.

The authors assert that "Variation, Location, State, and Interval are all extensible abstract base classes". However, Figure 2 does not contain a reference to the State class. Instead has an orange SequenceExpression class, which is not mentioned in the text but is perhaps the missing base class.

VRS has gone through several revisions, and this text referenced an earlier draft state. Apologies for the mixup, this has been clarified in text.

Figure 3A is difficult to understand. The three examples Insertion, Deletion and Substitution are completely identical, with the exception of coloring inter-residue 6 in the Insertion, and 5, 6 and 7 in the other two. How the two residue notations differ in the resulting description is not described. Figure 3B appears to be an example based on 3A, but the logic of coordinate assignment is not clear.

We have redesigned Figure 3A to better illustrate the conceptual difference between residue and inter-residue coordinates. Figure 3B has been redesigned to better convey that this is a depiction of different variant representation formats for the same variant. We understand the confusion from this figure, and think that our use of the "coordinate" column caused Figure 3B to appear to be an extension of the coordinate discussion in 3A. Among other design changes to this panel, we have removed the "coordinate" column and replaced it with a "description" column instead.

The description of a "globally unique identifier" lacks a rationale about the advantages of such a concept and why such thing is valuable beyond the canonical JSON. The detailed technical algorithm may be beyond relevance for readers of Cell Genomics. It is also not clear to this reviewer how different sequence identifiers lead to an identical VRS identifier, unless they are resolved to their underlying sequences.

Variants are commonly described using compact descriptive nomenclatures such as HGVS, which relies upon sequence identifiers such as RefSeq or Gencode accessions as context for the variant allele. In some systems, variants may also be represented by a registered identifier, such as ClinVar, CIViC, or ClinGen Allele Registry IDs. The many different forms by which a variant is referenced creates challenges in interoperability by relying upon pre-negotiated use of reference sequences or shared registries, forcing reliance upon centralized resources. VRS computed identifiers provide a method for constructing an identifier that is not reliant upon a central authority; a VRS allele (and its identifier) may be constructed entirely from a reference sequence and the state of a location on that sequence. This means that two different identifiers for the same reference sequence resolve to the same computed identifier, reducing the need for prenegotiation of shared authoritative identifiers between systems. Figure 4B demonstrates the degeneracy of sets of *identical* variation that appear to be distinct due to

choices of sequence identifiers. The reviewer has also correctly intuited that tooling needs to exist to retrieve reference sequences from existing sequence identifiers to compute digests (or in some implementations, directly map sequence identifiers to precomputed digests). This may be handled by implementations of, for example, the previously published RefGet standard. The VRS-python package described in the manuscript uses the seqrepo library to do these lookups on NCBI and Ensembl sequences, as these represent the most common use cases. Please see https://vrs.ga4gh.org/en/latest/impl-guide/required_data.html for an elaboration of the data required for implementations.

We added a brief description of the motivation for computed identifiers in the final paragraph of the introduction.

Critique of the model

The authors do not discuss the ambiguity of the model. Unfortunately, one and the same variation can be described in two or more different ways without a mechanism of avoidance or disambiguation, leading to problems with processing by the computer. For example, a RepeatedSequence might easily be represented by a LiteralSequence. A substitution can be modeled as a combination of a deletion and insertion.

There are many sources of ambiguity when representing sequence variants. VRS eliminates some sources of ambiguity (e.g., human choice of sequence identifier), and minimizes others that result from conflicting requirements (e.g., some variation could be represented as repeats or as indel variation, but eliminating this ambiguity would entail unacceptable restrictions for repeat sequences).

We have added text describing the roles of different forms of Sequence Expressions to the manuscript to help clarify when each is to be used. We have also added a discussion section to the VRS v1.2 specification that provides guidance on when each expression type should be used (<https://vrs.ga4gh.org/en/1.2/appendices/equivalence.html#using-sequence-expressions>). In the discussion, we also acknowledge and discuss how VRS is not a panacea for variation representation, specifically when it comes to the many related contexts of variation (e.g. GRCh37 genomic, GRCh38 genomic, transcript, protein contexts). This is analogous to the issue of when a sequence should be represented as a RepeatedSequence (due to clinical significance or detection method for the repeated subunit) or a LiteralSequence (the context-agnostic representation that represents a literal state). However, we do provide guidance on how and when to use different variation structures in VRS, and refer to our developing originating context policy and value object descriptors in the discussion. In response to this reviewer's comments, we have added a link to these developing products as part of the VRS Added Tools for Interoperable Loquacious Exchange (VRSATILE) framework.

Reviewer #3:

In their manuscript, "The GA4GH Variation Representation Specification (VRS): A Computational Framework for the Precise Representation and Federated Identification of Molecular Variation", the authors present the key elements of VRS. These include the scope, a terminology/information model, a JSON schema, recommended conventions, globally unique computed identifiers, and tools to facilitate community adoption. The authors emphasize that VRS is intended as a computational framework offering precision, expressiveness, and extensibility in variant representation, rather than (another) human readable variation format. They suggest that VRS complements these other formats.

VRS is distinguished by several features: (1) the use of value objects, (2) distinction for the concepts of molecular and systemic variation, (3) use of inter-residue coordinates for specifying sequence locations, and (4) globally unique computed identifiers (suitable for use by other GA4GH specifications).

The primary strength of VRS is in its ability to promote accurate data exchange of variants- because it provides unique identifiers for variant objects, it should be possible to share data about variants regardless of their origin and/or any local or repository-specific identifiers.

It is notable that the authors of this manuscript hail from some of the largest and well-known institutions that generate, store, and evaluate variant data, and have an international distribution. This is meaningful as it suggests the existence of broad support for VRS, improving the likelihood of adoption. Provisions to promote adoptability, such as a translatable schema and packages that translate VRS to existing formats (HGVS and SPDI) should also help in this area. However, it still remains to be seen whether bioinformatics approaches to variant analysis have become sufficiently pervasive to that it will result in this computationally-focused specification becoming the lingua franca for variation.

The manuscript is well-written, as is the accompanying online documentation at <https://vrs.ga4gh.org/en/stable/index.html>.

I support the publication of this manuscript, with no substantial revisions. Several minor items for the authors to address are listed below.

Thank you for your support of our manuscript and VRS!

1. Figure 2 Information Model and the schema on the website (<https://vrs.ga4gh.org/en/stable/schema.html>) have some differences. Can those be harmonized?

Yes. We have updated the version 1.2 release candidate documentation in response to reviewer comments, and the updated specification documentation can be found at <https://vrs.ga4gh.org/en/1.2/>.

2. Section on implementation and community adoption: The text mentions that VRS has been adopted by and evaluated by several major institutions in the bioinformatics community (and then lists several by name). Can they more clearly distinguish who has adopted vs. who is evaluating at this time?

Supplementary table 2 lists all committed adopters, with the status of their adoption and implementation details provided where available; this table does not include information on groups that are still undecided / evaluating adoption.

3. VRS allele normalization (full justification) requires expanding boundaries to find non-ambiguous boundaries. Within the human genome, there are some regions of multi-megabase length repeats (and with the recent sequencing of human centromeres and telomeres, we can expect to see more variation reported in large repeats). What, if any, impact might these very large normalized allele representations have on storage and performance?

We also recognize that there are pragmatic limitations to LiteralSequenceExpressions, and in the event there was a need to precisely represent extremely large sequences in a literal form we would need to develop a way to do this differently. However, VRS also provides additional mechanisms for these types of challenges. For example, RepeatedSequenceExpressions may be used for describing repeated sequences by subsequence count, and DerivedSequenceExpressions may be used to reference large

regions of “approximately reference” sequence by location. We have added text to the manuscript describing these features.

4. If normalization is so strongly encouraged, why isn't it required?

This was a challenging design choice, and like many such choices in VRS and other specifications, represents our judgement about the benefit of strictly requiring a particular convention versus the (sometimes) hypothetical value of flexibility. We have cataloged many of these decisions and the rationale for them in https://vrs.ga4gh.org/en/1.2/appendices/design_decisions.html. In this case, we believe that there are corner cases in which requiring normalization might have unintended consequences. For example, normalization on a transcript sequence, which is unaware of exon structure, might cause a variant to shuffle across an intron boundary. While this might be appropriate for most cases, requiring normalization might prohibit adoption by implementers who have other needs. We wrote the normalization rules with the most common cases in mind, and expect that in most implementations they should and will be followed. We are also producing a framework to guide adoption and use of VRS in existing systems (the VRSATILE framework; vrsatile.readthedocs.io). We added text to the manuscript describing VRSATILE, as well as our rationale for recommending (instead of requiring) normalization.

5. The Computed VRS Identifiers provide unique identifiers for variation on the same molecule. Does the model provide any mechanisms for indicating equivalence or non-equivalence of variants on different molecules? (e.g. transcript to genome or genome version 1 to genome version 2)?

No. This is out of scope for VRS, but we recognize the importance of this concept, and are developing mechanisms to address this with Variation Concept Origination Policies (VCOPs) in VRSATILE.

Referee reports, second round of review

Reviewer #1: Thanks for the clarifications and improvements. I only have a few minor nitpicks which I trust you to address without a direct response.

Typo: "While the design choices ... enables" -- "enables" should be "enable"

Figure 2 (and the diagram in <https://vrs.ga4gh.org/en/1.2/schema.html#overview>): I see SimpleInterval has been moved down to the deprecated section of vrs.yaml so perhaps it should be omitted? More importantly, CytobandInterval has "integer" as the type of start and end. I think that should be "string" for simplicity. (I see in vrs.yaml that the actual type of start and end is HumanCytoband, but that's just string with a regex spec, no need to clutter up the diagram with that.) CytobandInterval is also missing "+ type: string", making it look like the odd one out in the General Purpose Types panel, but from vrs.yaml it looks like it does include type ("CytobandInterval").

Figure 3C VRS Allele Normalization Algorithm: this is a great illustration! However, the "CA" that is trimmed from the sequences shown in step 1 disappears in step 2 -- I think it should be moved to either the left or the right of the brackets. It is present in the reference sequence in step 4, but then it is missing from the first two parts of step5. Then it reappears in the third part of step5, to the right of the insertion point. So I think it should be added back to step 2 and the first two parts of step 5, to the right of the brackets ("GCAGCT"  "CAGCAGCT").

Step 2 would look like this:

TCAG [/ CAG] CAGCAGCT

Step 5 would look like this:

Start

TCAG [/ CAG] CAGCAGCT

Expand to left bound

T [CAG / CAGCAG] CAGCAGCT

Reviewer #2: The quality of the revised paper has significantly increased, and the paper now clearly delineates the purpose and all the details of the proposed standard. However, the paper is still tough food for your typical molecular biologist and typical reader of this journal. I would therefore add some kind of a concise summary:

- What does the standard cover
- What does the standard not cover, why not, and why that is a good thing
- How it will be used in practice, and why is that important

Reviewer #3: The authors have sufficiently addressed the concerns raised by this reviewer; in addition they have made several other text and figure revisions that strengthen the overall readability of the manuscript and provide additional details/insight into the development of the model. Table 1, and the new section "Technical foundations and design decisions" are particularly welcome additions.

With regards to the now mentioned "RepeatedSequenceExpressions" and "DerivedSequenceExpressions", these are useful concepts that may help alleviate the need for LiteralSequenceExpressions in certain- but it is still not clear that they will be sufficient, and the authors note that new developments could still be needed. While this shouldn't be a blocker for publication, it should be noted that if such as need arises sooner than later, it could slow or hinder adoption of this model.

Minor points:

1. Improve the text for Supp Fig 2. There should be more explicit descriptions for each set of bricks. Additionally, define the LSA acronym in the figure text (e.g. ...three literal sequence Alleles (LSA; orange bricks)), as this acronym features significantly in the figure.

Author response to the second round of review

Reviewers' Comments:

Reviewer #1: Thanks for the clarifications and improvements. I only have a few minor nitpicks which I trust you to address without a direct response.

Typo: "While the design choices ... enables" -- "enables" should be "enable"

We have fixed this typo.

Figure 2 (and the diagram in <https://vrs.ga4gh.org/en/1.2/schema.html#overview>): I see SimpleInterval has been moved down to the deprecated section of vrs.yaml so perhaps it should be omitted? More importantly, CytobandInterval has "integer" as the type of start and end. I think that should be "string"

for simplicity. (I see in vrs.yaml that the actual type of start and end is HumanCytoband, but that's just string with a regex spec, no need to clutter up the diagram with that.) CytobandInterval is also missing "+ type: string", making it look like the odd one out in the General Purpose Types panel, but from vrs.yaml it looks like it does include type ("CytobandInterval").

We have made these revisions to Figure 2 and the online figure. We appreciate the careful eye for detail!

Figure 3C VRS Allele Normalization Algorithm: this is a great illustration! However, the "CA" that is trimmed from the sequences shown in step 1 disappears in step 2 -- I think it should be moved to either the left or the right of the brackets. It is present in the reference sequence in step 4, but then it is missing from the first two parts of step5. Then it reappears in the third part of step5, to the right of the insertion point. So I think it should be added back to step 2 and the first two parts of step 5, to the right of the brackets ("GCAGCT"  "CAGCAGCT").

Step 2 would look like this:

```
TCAG [ / CAG ] CAGCAGCT
```

Step 5 would look like this:

Start

```
TCAG [ / CAG ] CAGCAGCT
```

Expand to left bound

```
T [ CAG / CAGCAG ] CAGCAGCT
```

Great catch. We have made these revisions.

Reviewer #2: The quality of the revised paper has significantly increased, and the paper now clearly delineates the purpose and all the details of the proposed standard. However, the paper is still tough food for your typical molecular biologist and typical reader of this journal. I would therefore add some kind of a concise summary:

- What does the standard cover
- What does the standard not cover, why not, and why that is a good thing
- How it will be used in practice, and why is that important

We appreciate this suggestion and have added a "Highlights and eToc" component to the manuscript submission, summarizing these points for introduction to a more general audience. We have also expanded on this with slightly more detail as Box 1 in the manuscript introduction.

Reviewer #3: The authors have sufficiently addressed the concerns raised by this reviewer; in addition they have made several other text and figure revisions that strengthen the overall readability of the

manuscript and provide additional details/insight into the development of the model. Table 1, and the new section "Technical foundations and design decisions" are particularly welcome additions.

With regards to the now mentioned "RepeatedSequenceExpressions" and "DerivedSequenceExpressions", these are useful concepts that may help alleviate the need for LiteralSequenceExpressions in certain- but it is still not clear that they will be sufficient, and the authors note that new developments could still be needed. While this shouldn't be a blocker for publication, it should be noted that if such as need arises sooner than later, it could slow or hinder adoption of this model.

As with all components of VRS, we extend the model as needed to support community use cases. At present, we think these are sufficient, but have added the following text to make clear our philosophy of expanding the specification to meet community needs:

"Sequence Expressions are extensible and may be expanded in the future to other forms as needed to support community cases."

Minor points:

1. Improve the text for Supp Fig 2. There should be more explicit descriptions for each set of bricks. Additionally, define the LSA acronym in the figure text (e.g. ...three literal sequence Alleles (LSA; orange bricks)), as this acronym features significantly in the figure.

We have added explicit descriptions to the remaining bricks shown in Figure S2, and added the LSA acronym as suggested.